# Induction of Proteasome Subunit Low Molecular Weight Protein (LMP)-2 Is Required to Induce Active Remodeling in Adult Rat Ventricular Cardiomyocytes

**DOI:** 10.3390/medsci8020021

**Published:** 2020-05-01

**Authors:** Antonia Petersen, Hanna Sarah Kutsche, Franziska Nippert, Rolf Schreckenberg, Rainer Schulz, Klaus-Dieter Schlüter

**Affiliations:** Department of Physiology, Justus-Liebig-University, 35392 Giessen, Germany; antoniapetersen@gmx.de (A.P.); Hanna.Kutsche@physiologie.med.uni-giessen.de (H.S.K.); bergmann.franziska@yahoo.com (F.N.); Rolf.Schreckenberg@physiologie.med.uni-giessen.de (R.S.); Rainer.Schulz@physiologie.med.uni-giessen.de (R.S.)

**Keywords:** cardiac remodeling, cardiac hypertrophy, proteasome

## Abstract

Isolated adult rat ventricular cardiomyocytes (ARVC) adapt to the two-dimensional surface of culture dishes once they are isolated from the three-dimensional heart tissue. This process mimics aspects of cardiac adaptation to pressure overload and requires an initial breakdown of sarcomeric structures. The present study therefore aimed to identify key steps in this remodeling process. ARVC were cultured under serum-free or serum-supplemented conditions and their sizes and shapes were analyzed as well as apoptosis and the ability to disintegrate their sarcomeres. ARVC require serum-factors in order to adapt to cell culture conditions. More ARVC survived if they were able to breakdown their sarcomeres and mononucleated ARVC, which were smaller than binucleated ARVC, had a better chance to adapt. During the early phase of adaptation, proteasome subunit low molecular weight protein (LMP)-2 was induced. Inhibition of LMP-2 up-regulation by siRNA attenuated the process of successful adaptation. In vivo, LMP-2 was induced in the left ventricle of spontaneously hypertensive rats during the early phase of adaptation to pressure overload. In conclusion, the data suggest that breakdown of pre-existing sarcomeres is optimized by induction of LMP-2 and that it is required for cardiac remodeling processes, for example, occurring during pressure overload.

## 1. Introduction

Isolated adult rat ventricular cardiomyocytes (ARVC) are able to adapt to the two-dimensional surface of a culture dish once they are isolated from the three-dimensional heart tissue [1,2]. During this process, cells rapidly degrade their sarcomeres and round down. Subsequently, they rebuild sarcomeres along pseudopodia-like structures. The latter process of the rebuilding of sarcomeres has received some attention, as this resembles processes that also occur during cardiac adaptation to stressors such as ischemia or pressure overload. Key players that participate in this process, such as swiprosin and oncostatin M, are up-regulated during adaptation in vitro and in vivo, and are causally linked to these processes [2,3]. The initial steps of cardiac adaptation to pressure overload have attracted less attention. However, degradation of sarcomeres is a prerequisite to perform active adaptation processes and reformation of sarcomeres. Therefore, the following study was performed to investigate specifically the initiation process of cardiac adaptation in vitro. 

ARVC display a heterogeneous cell population of mono and binucleated cells [4,5,6]. The latter ones are normally bigger (more hypertrophic) and may be considered as more differentiated. The ratio of mono to binucleated cells in the heart is species-specific. In the adult mouse heart, nearly all cells display a bi or polynucleated character [7], whereas in the adult human heart mononucleated cells are dominant but replaced by binucleated cells with ageing [8]. In the adult Wistar rat heart, binucleated cells dominate but mononucleated cells can be found as well [9,10,11]. Polyploidization of adult cardiomyocytes is the result of the inability of cardiomyocytes to complete cytokinesis [12]. Individual cardiomyocytes may contain nuclei with diploid, tetraploid, or octaploid DNA content [13]. Mono and binucleated cells from the left atrium have different electrical activity and calcium dynamics but a similar transcriptome [14,15]. It is currently completely unknown whether mono and binucleated ventricular cells differ in their structural flexibility, as it is required to adapt to cell culture conditions or to react to altered pressure load. Cellular remodeling can be analyzed by cultivation of ARVC, which requires adaptation to a two-dimensional surface of a cell culture dish. A difference in the ability to adapt to new conditions may be relevant in hearts at advanced age with more binucleated cells. It may indicate a reduced availability of the heart to respond to stress with adaptive remodeling [16]. 

ARVC adapt to culture conditions by the formation of pseudopodia-like structures [2,17,18,19,20]. However, whereas the formation of pseudopodia-like structures in cultured ARVC is an active process requiring excessive substitution of growth factors from serum, it is still unclear whether the initiation of this process, i.e., the degradation of preexisting sarcomeres, also requires the stimulation by growth factors. Alternatively, it may be that mechanical unloading under culture conditions initiates sarcomere degradation and growth factors stimulate the second phase, namely the formation of pseudopodia-like structures. 

In light of the present study, we therefore aimed to elucidate (1) whether degradation of sarcomeres during cultivation is an active process requiring activation of growth-factor-dependent signal transduction pathways, (2) whether mono and binucleated cells behave in a similar way with respect to adaptation to culture conditions, and (3) key molecules that actively trigger the process of sarcomere degradation and to investigate subsequently whether these key molecules are actively regulated under conditions of pressure overload that induce cardiac hypertrophy in vivo. We identified proteasome subunit low molecular weight protein (LMP)-2 as a key player in the earliest cellular adaptation of ARVC to remodeling and confirmed its induction under pressure overload in vivo. LMP-2 was initially considered as a molecular trigger of inflammatory response as it occurs during myocarditis [21]. Meanwhile, it is evident that it is also relevant for sterile inflammation as part of cardiac hypertrophy [22]. LMP-2 replaces the β1 subunit of a proteasome reducing caspase-like activity of a proteasome to background activity [23]. Via this process, LMP-2 contributes to an increased protein degradation of specific proteins [24]. The data are therefore relevant for our understanding of cardiac remodeling. 

## 2. Materials and Methods 

### 2.1. Cell Isolation and Cultivation

The investigation was conducted according to the Guide for the Care and Use of Laboratory Animals published by the US National Institute of Health (NIH Publication No. 85-23, revised 1996). The protocol was approved by the Justus-Liebig University Gießen (permission number: 561_M). Sacrifice was performed under isoflurane anesthesia and all efforts were made to minimize suffering of the animals. 

Ventricular cardiomyocytes of four month old male Wistar rats were isolated as described previously [25]. Briefly, the rat was sacrificed by cervical dislocation under deep anesthesia with isoflurane. The heart was immediately transferred into ice-cold saline solution. It was fixated on the cannula of a Langendorff perfusion system, followed by a 25 min lasting perfusion with a Powell medium (50 mL) containing collagenase (25 mg) and CaCl_2_ (25 µM) at 37 °C. Subsequently, ventricular tissue was minced and incubated in 5 mL of the recirculated buffer for five minutes. The remaining cell solution was filtered through a nylon mesh (200 µm). After centrifugation and stepwise addition of CaCl_2_ (200, 400 and 1000 µM), cells were plated on petri dishes coated with 4% vol/vol (fetal calf serum (FCS). 

Cultivation of the cells was performed as previously described [26]. Briefly, medium 1999 was supplemented with carnitine (2 mM), creatine (5 mM) and taurine (5 mM), penicillin-streptomycin (2%) and 20% FCS. Where indicated cells were cultured in the absence of 20% FCS, Cytosine-β-arabinofuranoside (10 µM) was supplemented to hamper the proliferation of any contaminating cells. The incubator was held at 37 °C. 

With each cell preparation, cardiomyocytes were evaluated per day by light microscopy. All counted cardiomyocytes were subdivided into “rod-shaped” or “round down” cells, as explained before [2]. 

In order to analyze the morphological and structural adaptation of ARVC in culture, confocal laser scan microscopy was performed. Phalloidin TRITC was used to stain F-actin according to the manufacturer’s protocol (Santa Cruz Biotechnology, Heidelberg, Germany).

### 2.2. Spontaneously Hypertensive Rats

Tissue samples to analyze the expression of LMP-2 in left ventricular tissue from spontaneously hypertensive rats were depicted from a previous study, in which these samples were used as age-matched controls [27]. Details about the measurements of blood pressure and sample storage can also be depicted from this reference.

### 2.3. RT-PCR

Real-time quantitative RT-PCR in ARVC was performed as described initially in reference [28]. Total RNA was isolated using peqGOLD TriFast according to the manufacturer’s protocol (Peqlab, Biotechnologie GmbH, Erlangen, Germany). After conversion of RNA into complementary DNA (cDNA) with reverse transcriptase, PCR was performed with iQ^TM^SYBR^®^ Green Supermix (Bio Rad, Düsseldorf, Germany). In order to detect unspecific binding, melting curves were analyzed. For experiments with RT-PCR, ARVC of four petri dishes were combined. The primers used are listed in the Appendix A.

### 2.4. Western Blot

Isolated ARVC were incubated with lysis buffer as described before [28]. SDS-PAGE gel electrophoresis was conducted with the system of NuPAGE, Novex^®^ (Life Technologies, Carlsbad, CA, USA). A list of all antibodies used in this study is given in the Appendix A. Protein expression was quantified with horseradish peroxidase and a chemiluminescence analyzer from Peqlab (Erlangen, Germany). Data on protein expression are always normalized to GAPDH. 

### 2.5. Statistics

Statistical analysis was performed using SPSS 22. Data are expressed as indicated in the legends. ANOVA and the Student–Newman–Keuls test for post hoc analysis were used to analyze experiments in which more than one group was compared. Normal distribution of the samples was determined by the Shapiro–Wilk test and equality of variance was estimated by the Levene test. Data of two samples were analyzed by *t*-test, Welch’s test, or the Mann–Whitney test. *p* < 0.05 was marked as significant.

## 3. Results

### 3.1. Kinetics of Sarcomere Degradation

ARVC do not directly adapt to culture dishes in their rod-shaped cell form. They pass through a two-fold process, by which sarcomeres are first degraded and subsequently reformed. Sarcomere degradation converts the rod-shaped cell morphology into round cells. Figure 1 illustrates this process. 

Freshly isolated ARVC (Figure 1A) displayed either one or two elongated nuclei and a rod-shaped phenotype. Forty-eight hours later, cells that successfully adapted to the culture dish rounded down (Figure 1B(a)). Some cells behaved differently, as they still rounded down but displayed DNA condensation (round nuclei) (Figure 1B(b)). Finally, some cells were still rod-shaped (Figure 1B(c)). Figure 2 shows a representative phalloidin staining of freshly isolated rod-shaped ARVC (Figure 2A). Within the first 48 h after cultivation, ARVC started to degrade their sarcomeres, beginning from the cell poles. As a consequence of this, cells displayed a round instead of rod-shaped cell surface (Figure 2B). In Figure 2C, the amount of cells with a round cell shape was quantified at day 0 (day of cell isolation) and day 2. In the presence of 20% FCS, approximately 70% of the cells round down, whereas without FCS they remain rod-shaped. In the video uploaded as Appendix A, it can be seen that indeed rod-shaped cells undergo the morphological differentiation (Appendix A).

### 3.2. Cell Shape of Mono and Binucleated Cells and Adaptation to Culture Conditions

Mononucleated and binucleated ARVC significantly differed in length and volume (Figure 3A–C). As a consequence of this, the nuclei-to-volume ratio remained similar in both types of myocytes. Moreover, whereas more than 90% of mononucleated cells successfully rounded down and adapted to culture conditions, less than 60% of binucleated cell did so (Figure 3D). 

Consequently, after two days, nearly 30% of all cells were mononucleated cells (Figure 4A). This is mainly due to apoptotic cell death of binucleated cells, whereas mononucleated cells remained viable (Figure 4B). In the absence of FCS, mono and binucleated cells undergo apoptotic cell death (Figure 4B).

### 3.3. Molecular Adaptation of ARVC during Cultivation

Cultivation of ARVC in serum-containing medium induces morphological adaptation to the culture dish. This was accompanied by molecular changes in the expression of cardiomyocyte specific proteins (Figure 5A). The mRNA expression of sarcomeric proteins like troponin T (TnT), α myosin heavy chain, and β myosin heavy chain was reduced, while that of the stress fiber protein β-actin was increased. Furthermore, LMP-2 was induced. LMP-2 is an inducible β subunit of the proteasome that is associated with an increased substrate turnover. Furthermore, serum cultivation induced the mRNA expression of stromal cell-derived factor (SDF) and CXCR4, the receptor for SDF. Bax, a pro-apoptotic protein, is also induced. The adaptation of ARVC to culture conditions requires protein degradation related to the induction of LMP-2 and reduction of the formation of sarcomeric proteins, like β-MHC. Transcriptional changes in the expression of these proteins was confirmed on the protein level, increased protein expression for LMP-2 (Figure 5B,C) and reduction of the sarcomere protein β-MHC (Figure 5B,C). The stress-fiber protein β-actin was not significantly induced at this time point.

### 3.4. Activation of LMP-2 and Contribution to Cellular Adaptation to Culture Conditions

LMP-2 exerts its function when it is incorporated into the proteasome thereby replacing conventional β isoforms of the proteasome complex. During this process, it is transferred into a smaller subunit. As the activation of the proteasome can be considered as a stress response, we investigated whether inhibition of the activation of the stress-activated p38 mitogen activated protein (MAP) kinase can attenuate the activation of LMP-2. This was indeed the case, because SB202190, a specific inhibitor of p38 MAP kinase-dependent translocation of myocyte enhancer factor 2C into the nucleus [29], attenuated the induction of LMP-2 (Figure 6A,B), its activation (Figure 6C), and finally also the process of cellular adaptation to culture conditions (Figure 6D).

As the p38 MAP kinase inhibitor may exert unspecific effects, we finally investigated whether silencing of LMP-2 is sufficient to impair cellular adaptation to culture conditions. As indicated in Figure 7A, siRNA directed against LMP2 significantly attenuated LMP-2 protein expression (Figure 7A,B), but also cellular adaptation (Figure 7C). 

### 3.5. Induction of LMP-2 during the Adaptation of the Left Ventricle to Pressure Overload

Finally, we investigated the expression of LMP-2 in a model of cardiac hypertrophy. Spontaneously hypertensive rats (SHR) were analyzed at the age of 1.5 (pre-hypertensive state) to 7.0 months (established hypertension). Blood pressure increased between 1.5 and 3.0 months (Figure 8A). At the same time, heart weight to body weight increased (Figure 8B). Atrial natriuretic peptide (ANP), a molecular marker of hypertrophy, increased in parallel (Figure 8C). LMP-2 mRNA expression increased during the early phase of hypertrophy but was normalized at later time points when cardiac hypertrophy was fully established (Figure 8D).

## 4. Discussion

The main finding of our study is the observation that LMP-2 is induced early during adaptation of ARVC to cell culture conditions. This process requires an activation of a p38 MAP kinase-dependent pathway. Without this early activation of p38 MAP kinase and induction of LMP-2, ARVC are unable to survive under culture conditions. Moreover, a similar induction of LMP-2 characterizes the early phase of adaptation to pressure overload, supporting the hypothesis that adaptation of ARVC to cell culture conditions mimics key aspects of cardiac adaptation to pressure overload. 

Cardiac remodeling is a complex process including intracellular reconstruction in myocytes [30]. Cultivation of ARVC in serum-supplemented media allows us to analyze these processes in more detail. In the past, a focus has been set on the reformation of sarcomeres, while the initiation of this process, namely the degradation of existing sarcomeres, has achieved less attention. We have previously shown that during the cultivation process, an increased expression of p38 MAP kinase occurs [31]. Activation of p38 MAP kinase drives embryonic stem cells into cardiomyocyte-specific differentiation by enhancing the transcription of myocyte enhancer factor 2C, a transcription factor required for cell differentiation [32,33]. Our finding of p38 MAP kinase-dependent effects on cell adaptation may indicate that the activation of similar signal transduction pathways is associated with remodeling processes in adult cardiomyocytes as well. Furthermore, an imbalance between protein degradation and protein synthesis (atrophy) is a challenge for an individual cell, leading to apoptosis [34,35]. Therefore, specific mechanisms are required that allow ARVC to adapt to altered requirements due to changes in growth factor composition and wall tension. In vivo studies suggest that an activation of the proteasome contributes to this type of adaptation [36,37,38,39]. Here, we provide evidence that an induction of LMP-2 is part of this process. LMP-2 is incorporated into the proteasome apparatus, leading to accelerated proteolytic activity [40]. Increasing the capacity of proteasome activity is necessary to avoid an accumulation of proteins in the cell. This holds specifically for those proteins that are formed by protein carbonylation as it occurs during oxidative stress [35]. Our finding is based on the following observations: First, under culture conditions, LMP-2 expression is increased prior to cell rounding (which means degradation of sarcomeres building the rod-shaped character). Second, LMP-2 is found in its proteolytic form as it is required for incorporation into the proteasome. Third, silencing of LMP-2 by administration of siRNA directed against LMP-2 is sufficient to reduce the number of cells able to adapt to cell culture conditions. Finally, LMP-2 is induced during the early adaptive phase of myocardial hypertrophy in SHR, indicating that LMP-2 induction is not only a necessary factor for adaptation of ARVC to culture conditions but in a more general view required for cardiac remodeling in vivo. In the same animal model, increased proteasome activity has been linked to adaptive hypertrophy [35]. In line with these assumptions, hearts from LMP-2 knockout mice displayed massive hypertrophy and decreased fractional shortening, indicating maladaptive hypertrophy in the absence of LMP-2 [22]. 

Sarcomere degradation may be considered as a reaction to mechanical unloading as it occurs during cultivation. Without connection to a conduction system and without electrical pacing, cardiomyocytes remain quiescent. However, if mechanical unloading would trigger sarcomere degradation, cell rounding would appear under serum-free and serum-supplemented conditions. This is, however, not the case. Therefore, we consider the mix of growth factors present in serum as a trigger for this process. As mentioned before, p38 MAP kinase is induced during the cultivation process. Therefore, it is likely to assume that p38 MAP kinase participates in this process. This was indeed the case as a pharmacological inhibition of the kinase attenuated the process of adaptation. As for LMP-2, p38 MAP kinase activation is part of the signaling in cardiac remodeling in SHRs. Moreover, p38 MAP kinase is already induced at the adaptive phase of hypertrophy. In conclusion, induction of LMP-2 is simultaneously found in culture adaptation of ARVC and in SHRs during the adaptation to pressure overload. 

ARVC are large cells that display, depending on their size, one or two and eventually more than two cell nuclei. In the rat heart, ARVC have approximately 10–15% mononucleated cells. Overall, there seems to be a fixed ratio between the size of the cell and the number of nuclei. This means that binucleated cells are larger. Here, we investigated whether these cells adapt to culture conditions in a different way. It was found that mono and binucleated cells adapt to culture conditions. However, the amount of cells that successfully adapt is higher in the group of mononucleated cells. As the amount of binucleated cells increases during ageing, the consequence of this would be that during the ageing process, the ability of the heart to adapt to pressure overload decreases. Unfortunately, afterload increases during ageing, thereby producing an unfavorable situation in which more cells within the heart need to be reconstructed but fewer cells have the ability to do so. This may contribute to the long-term maladaptation of elder hearts. Another aspect not determined in this study is that the ratio of diploid to polyploid nuclei will change during ageing and also depends on species [41]. The impact of polyploidy on cell adaptation could not be analyzed in this study, but due to previous studies, will not be different in rat myocytes [41]. 

The current study deals exclusively with adult rat ventricular cardiomyocytes. As such, they represent fully differentiated cells that require extensive cellular remodeling for adaptation to culture dishes. The model allows us to monitor such a process in a detailed and accelerated way. However, the underlying mechanism may also be relevant to cardiac remodeling in a more general way. Although, comparable experiments with human cardiomyocytes have not yet been performed, a similar contribution of proteasome activation, including the incorporation of inducible proteasome subunits such as LMP-2 has been suggested in human heart failure; here, insufficient proteasome-dependent degradation of ubiquitinated proteins has been shown in cardiomyocytes in human heart failure [42]. Further support for the importance of proteasome function comes from animal models, in which the amelioration of proteasome function aggravates initial stages of cardiac disease [43]. Therefore, species-specific differences are not very likely to be assumed in this aspect. Pharmacological inhibition of proteasome activity is possible with MG-132. However, when used in our study, the drug was toxic for the cells (1–10 nM). When reducing the concentration to non-lethal concentrations of MG-132 (<1 nM), we did not observe an effect on spreading, but in this, case the concentration was below the Ki (4 nM) and probably not sufficient. It remained unclear for us, why MG-132 was successfully used by others in the same cell system (5 µM) [44]. However, in these experiments, MG-132 was used for 6 h only, whereas our experiments required 48 h incubations. Furthermore, Wang et al. performed single cell experiments (Patch Clamp). Individual cells may survive but this was not sufficient for our type of analysis.

## 5. Conclusions

Our study provides evidence that the induction of the proteasome activity is a necessary step during the initiation of adaptive hypertrophy. This step requires induction of LMP-2 and subsequent increasing in the capacity for protein degradation. This is necessary to avoid an accumulation of proteins within cells as it occurs in other chronic diseases as well. Furthermore, smaller mononucleated cells seemed to have a greater structural flexibility, allowing them better to adapt mechanical load. This seems to be important during the ageing process in which the number of mononucleated cells is getting smaller and that of polynucleated cells increases. Although this enables the cell to maintain a stable nuclear-cell volume ratio, it is associated with a loss of structural flexibility.

## Figures and Tables

**Figure 1 medsci-08-00021-f001:**
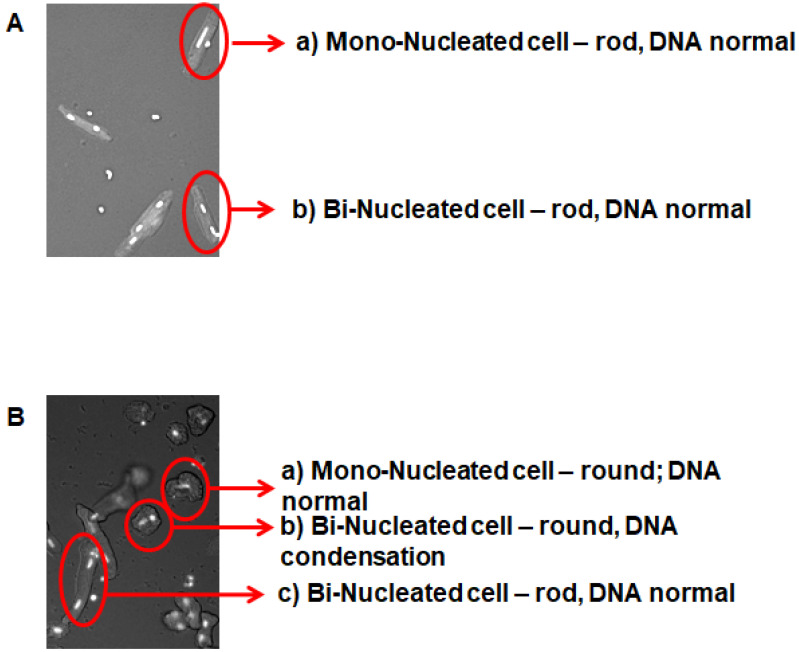
Representative pictures of adult rat ventricular cardiomyocytes. (**A**) Freshly isolated cardiomyocytes are shown; (**B**) cardiomyocytes after 48 h cultivation in the presence of fetal calf serum.

**Figure 2 medsci-08-00021-f002:**
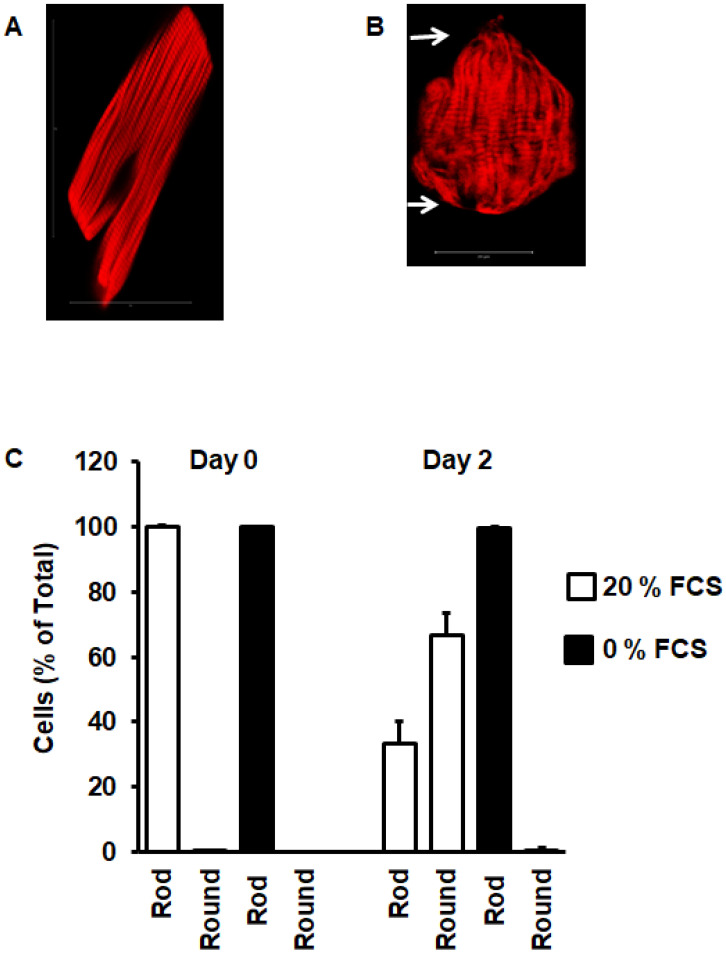
Cell rounding during cultivation. (**A**) Freshly isolated cardiomyocyte with phalloidin staining of actin to visualize striation of sarcomeres. (**B**) Cardiomyocytes after 48 h cultivation in the presence of fetal calf serum, indicating sarcomere degradation from the cell poles. (**C**) Quantification of the number of rod-shaped cardiomyocytes and round cardiomyocytes at start of cultivation (day 0) and at day 2. Data show the effect of serum on cell rounding. Data are means ± SD from *n* = 8 preparations (237–410 cells per preparation were analyzed).

**Figure 3 medsci-08-00021-f003:**
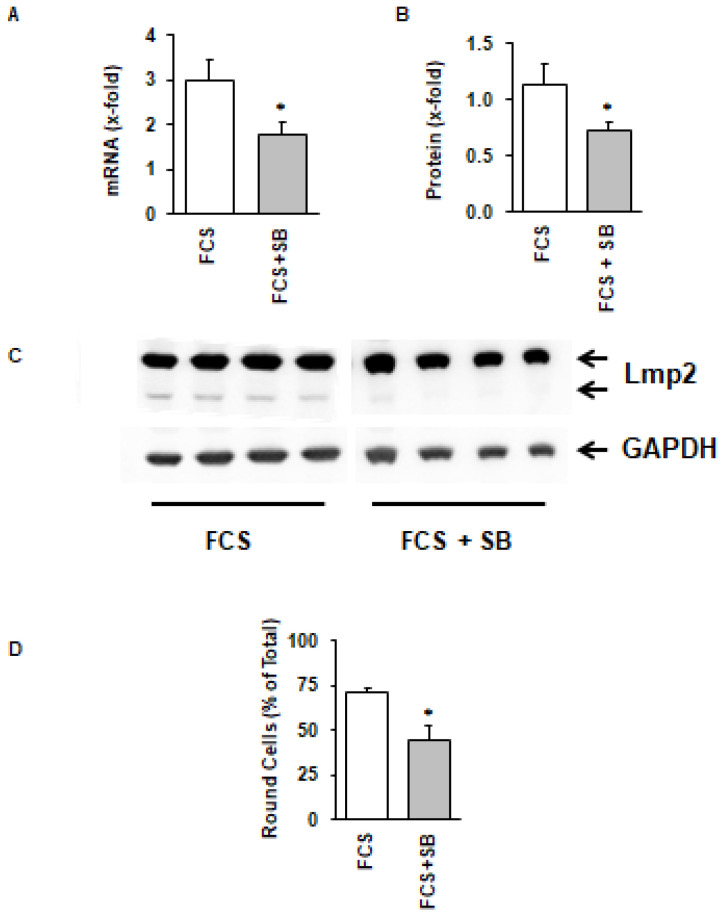
Cell size (**A**–**C**) of freshly isolated cardiomyocytes with one (*n* = 1) or two (*n* = 2) nuclei (**A**, µm length; **B**, µm width; **C**, µm^3^ volume). Percent of cells to round down in the presence of fetal calf serum is shown in (**D**). Data are means ± SD from *n* = 11 preparations (96–304 cells per preparation). (*, *p* < 0.05 vs. FCS)

**Figure 4 medsci-08-00021-f004:**
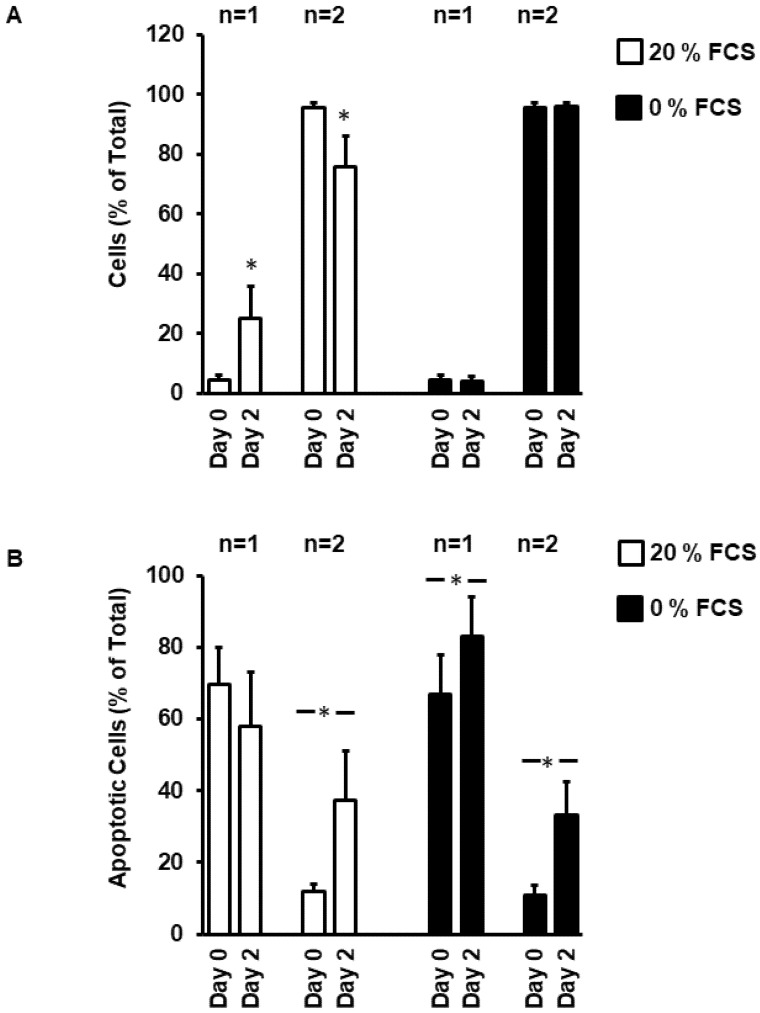
Ratio between mono (*n* = 1) and binucleated (*n* = 2) cells at day 0 and 2 in the presence and absence of fetal calf serum (**A**). Number of cells undergoing apoptosis (identified by DNA condensation—see Figure 1) under these conditions (**B**). Data are means ± SD from *n* = 8 preparations (237–410 cells per preparation). (*, *p* < 0.05 vs. Day 0)

**Figure 5 medsci-08-00021-f005:**
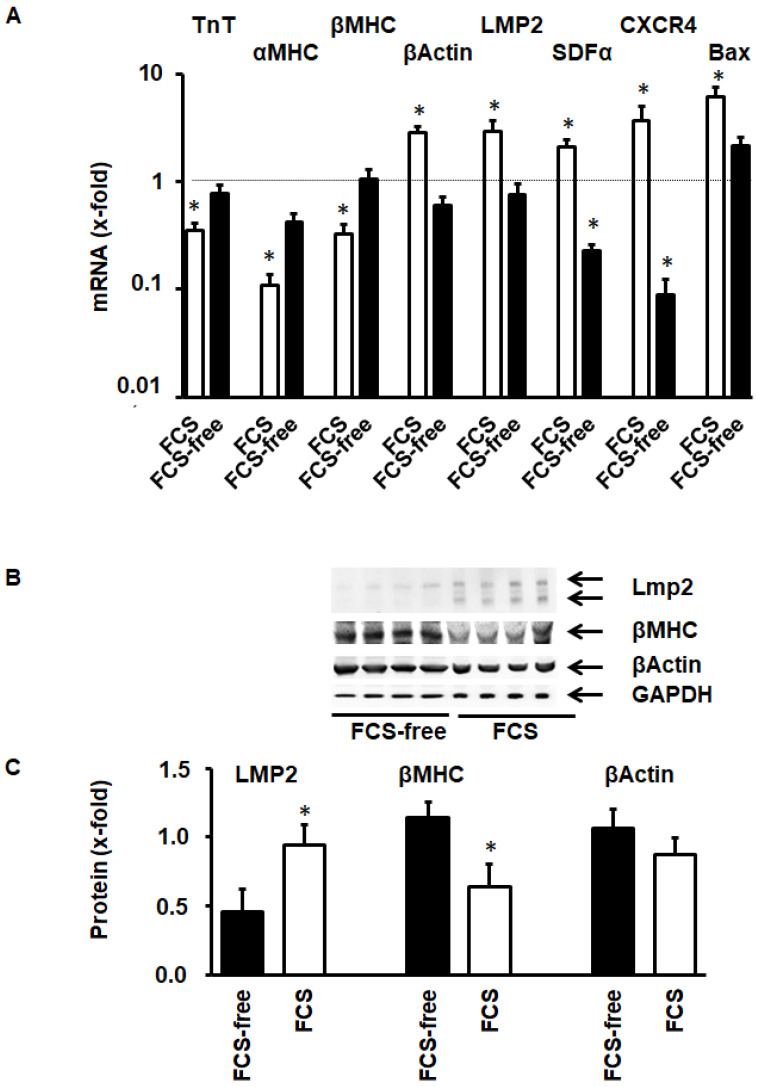
Protein expression in cultured cardiomyocytes within 48 h in the presence or absence of fetal calf serum (FCS). (**A**) mRNA expression of proteins of interest (*n* = 6). (**B**) Representative western blot of LMP-2, myosin heavy chain, and β-actin. The double band of LMP-2 in B indicates activation of LMP-2 and incorporation into proteasomes. (**C**) Quantitative analysis of the western blot shown in B. Data are means ± SD. (*, *p* < 0.05 vs. FCS free)

**Figure 6 medsci-08-00021-f006:**
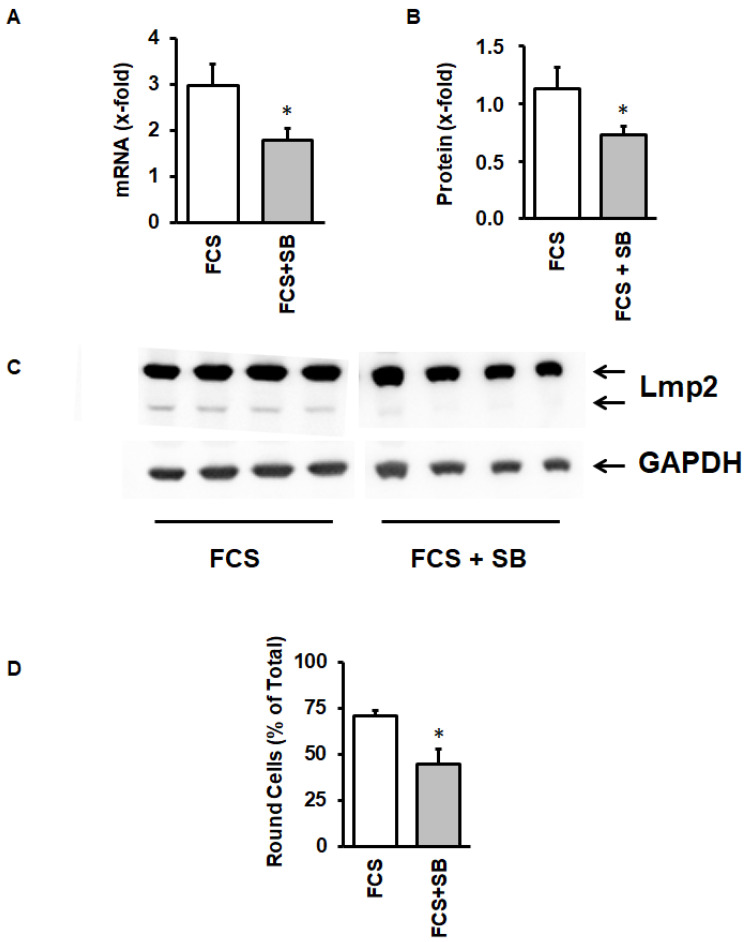
Effect of the p38 MAP kinase inhibitor SB202190 (SB) on LMP-2 mRNA expression (**A**), protein expression (**B**,**C**), and cell rounding (**D**) of cultured cardiomyocytes. Data are means ± SD from *n* = 4 preparations. *, *p* < 0.05 vs. FCS.

**Figure 7 medsci-08-00021-f007:**
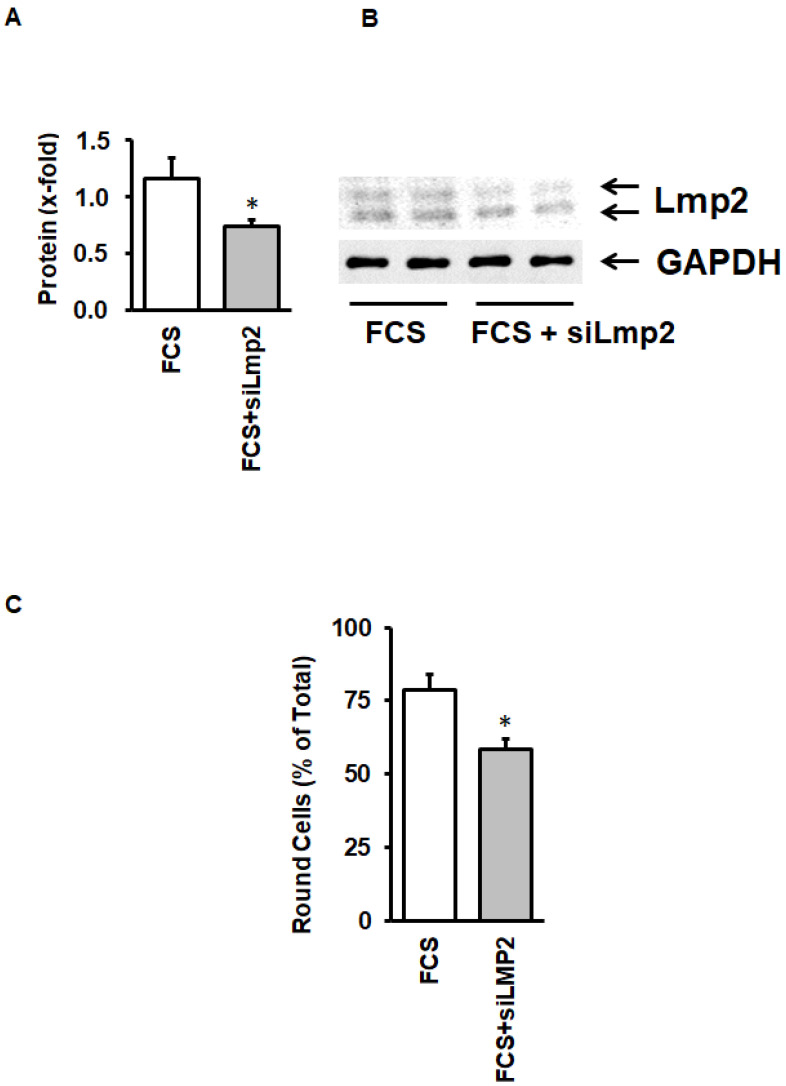
Effect of siRNA directed against LMP-2 on LMP-2 expression and cell rounding in cultured cardiomyocytes. (**A**) LMP-2 protein expression normalized to GAPDH; (**B**) Original Western Blot showing the expression of LMP-2; (**C**) Number of round cells after incubation with FCS or FCS and siRNA directed against LMP-2. Data are means ± SD from *n* = 5 preparations. *, *p* < 0.05 vs. FCS.

**Figure 8 medsci-08-00021-f008:**
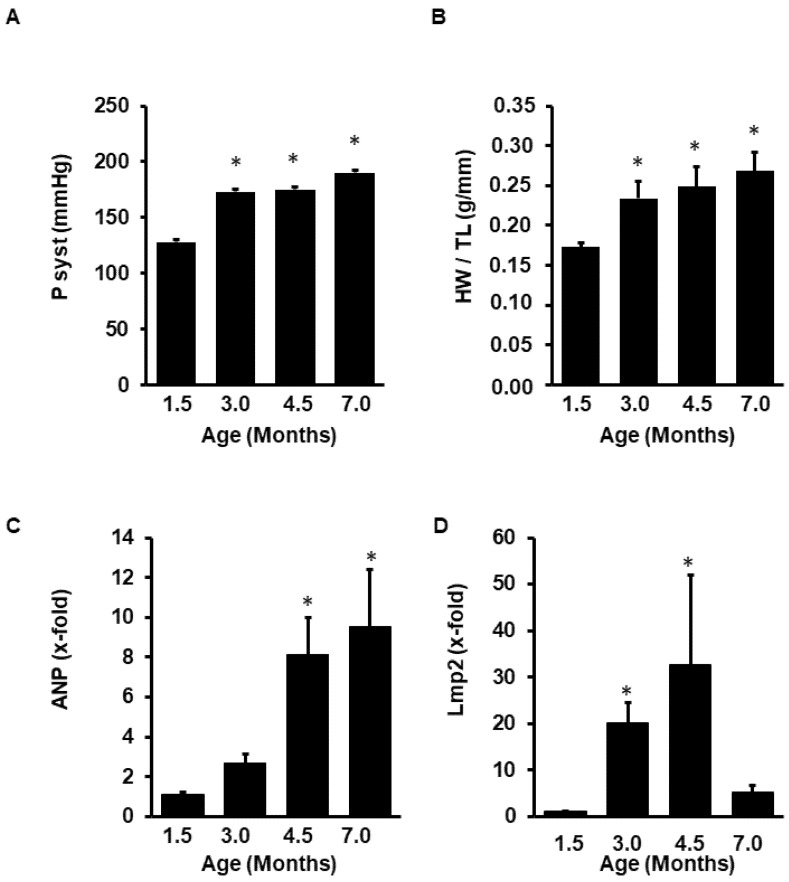
Expression of LMP-2 in spontaneously hypertensive rats. Data show the increase in blood pressure during ageing (**A**), the increase in heart weight to tibia length (HW/TL) as a parameter of hypertrophy (**B**), the induction of atrial natriuretic peptide (ANP; **C**), and the mRNA expression of LMP-2 (**D**) in the animals after 1.5–7.0 months as indicated. Data are means ± SD from *n* = 6 rats. *; *p* < 0.05 vs. 1.5 months.

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
