# Peer review of "Induction of Proteasome Subunit Low Molecular Weight Protein (LMP)-2 Is Required to Induce Active Remodeling in Adult Rat Ventricular Cardiomyocytes"

_medsci, 2020, doi:10.3390/medsci8020021_

Round 1

Reviewer 1 Report

The manuscript “Induction of proteasome subunit low molecular weight protein (LMP)-2 is required to induce active remodeling in cardiomyocytes” by Petersen et al. investigates the role of LMP-2 in rat cardiomyocytes during adaption to a cell culture environment.

The study is nicely conducted and the methods, results and conclusions are sound. Therefore the study might be of potential interest to the scientific community.

Still, I have a few minor comments that might improve the manuscript:

  • The manuscript investigates cardiomyocytes from rats only. This should be reflected in the title of the manuscript.

  • In the methods section line 100 it is stated that 150-3000 cardiomyocytes were evaluated. Are these numbers correct? Besides the fact that this section sounds a bit like copy and paste from the mentioned reference, Nippert et al. evaluated 150-300 cardiomyocytes.

“With each cell preparation 150 to 300 cardiomyocytes were evaluated per day and group by light microscopy. All counted cardiomyocytes were subdivided into four groups according to their appearance: “rod-shaped”, “round down”, “spreading” and “unusual appearance…”

  • In Figure 3C, 3D, 6A, 6B the Y-axis should be labelled properly.

  • The Discussion section could be strengthened if the authors would ad some comparison of rat cardiomyocytes to human cardiomyocytes.

Author Response

You wrote: The manuscript investigates cardiomyocytes from rats only. This should be reflected in the title of the manuscript.

Response: We have changed the title of the study. The new title reads: “Induction of proteasome subunit low molecular weight protein (LMP)-2 is required to induce active remodeling in adult rat ventricular cardiomyocytes”

You wrote: In the methods section line 100 it is stated that 150-3000 cardiomyocytes were evaluated. Are these numbers correct? Besides the fact that this section sounds a bit like copy and paste from the mentioned reference, Nippert et al. evaluated 150-300 cardiomyocytes.

Response: As indicated in the manuscript the method was adapted to the previous study by Nippert et al. We apologize for the mistake (3000 should read 300), removed the numbering of cells (150-300) in line 100 and give the exact numbering in figure legends 2-4 for clarification.

You wrote: In Figure 3C, 3D, 6A, 6B the Y-axis should be labelled properly.

Response: Thank you for your comment. We did not recognize that labeling in Figures 3 and 6 was partially lost but copy process from PowerPoint to the original manuscript file. The figures are changed according to your suggestion.

You wrote: The Discussion section could be strengthened if the authors would add some comparison of rat cardiomyocytes to human cardiomyocytes.

Response: Species independence of the observed effects is critical and requires further investigations. We have now included a statement that compares basal findings in animal and human tissues. Please read on line X: “The current study deals exclusively with adult rat ventricular cardiomyocytes. As such they represent fully differentiated cells that require extensive cellular remodeling for adaptation to culture dishes. The model allows us to monitor such a process in a detailed and accelerated way. However, the underlying mechanism may also be relevant to cardiac remodeling in a more general way. Although, comparable experiments with human cardiomyocytes have not yet been performed, a similar contribution of proteasome activation including the incorporation of inducible proteasome subunits such as lmp-2 have been suggested in human heart failure; here insufficient proteasome-dependent degradation of ubiquitinated proteins has been shown in cardiomyocytes in human heart failure [42]. Further support for the importance of proteasome function comes from animal models in which the amelioration of proteasome function aggravates initial stages of cardiac disease [43]. Therefore, species specific differences are not very likely to assume in this aspect.”

Reviewer 2 Report

The manuscript by Petersen et al investigated the role of LMP-2 in the induction of cardiomyocyte remodeling. The authors revealed that the induction of LMP2 is required for the induction of sarcomere breakdown in ARVCs. Inhibition of LMP-2 by siRNA or p38 inhibitor SB202190 prevented the adaptation of the ARVCs to culture conditions. The experimental design is straightforward and the conclusion is supported by their results. I have only a few comments for the authors to consider.

The authors claim that induction of proteasome activity is a necessary step during the initiation of adaptive hypertrophy and this step requires the induction of LMP-2. However, no direct evidence was shown that an increased proteasome activity accompanied by the increase of LMP-2 levels. Is there a cellular functional assay to reveal an increased proteasome activity during the adaption of the ARVCs to culture conditions?. Additionally, whether inhibition of the proteasome pathway has a similar effect of LMP-2 inhibition?

Line 197, a typo "Activatiopn" needs to be corrected.

Author Response

You wrote: The authors claim that induction of proteasome activity is a necessary step during the initiation of adaptive hypertrophy and this step requires the induction of LMP-2. However, no direct evidence was shown that an increased proteasome activity accompanied by the increase of LMP-2 levels. Is there a cellular functional assay to reveal an increased proteasome activity during the adaption of the ARVCs to culture conditions?

Response: The reviewer mentions a critical point namely the quantification of proteasome-dependent activity. In the light of the induction of Lmp-2, the best association would be an increased caspase-like activity. Indeed, in models of isoproterenol-induced heart failure a caspase-like activity is reduced (Drews et al., 2010). However, we did not perform such enzymatic experiments in the current study, but a pre-requisite for increased activity is the incorporation of Lmp-2 into proteasomes. This process depends on proteolytic activation of Lmp-2, not only on its expression. In this context please note that a characteristic double band is obtained only in FCS-treated cardiomyocytes (Fig. 5b) and this could be attenuated by co-administration of a p38 MAPK inhibitor (Fig. 6c). This strongly supports the suggestion that Lmp-2 is not only expressed but also incorporated into the proteasome complex and that the activity is also increased. In the absence of a direct assay that was not possible with the low amount of material from cell culture dishes this is the best assumption that can be given.

You wrote: Additionally, whether inhibition of the proteasome pathway has a similar effect of LMP-2 inhibition?

Response: We thank the reviewer for his/her comment. We worked on this previously but with the standard procedures described in the literature it was not possible to perform these studies. As the point is indeed of relevance we extended the discussion. Please read on line X: “Pharmacological inhibition of proteasome activity is possible with MG-132. However, when used in our study, the drug was toxic for the cells (1-10 nM). When reducing the concentration to non-lethal concentrations of MG-132 (<1 nM) we did not observe an effect on spreading, but in this case the concentration is below the Ki (4 nM) and properly not sufficient. It remained unclear for us, why MG-132 was successfully used by others in the same cell system (5 µM; ref. 44). However, in these experiments MG-132 was used for 6 h only whereas our experiments required 48 h incubations. Furthermore, Wang et al. performed single cell experiments (Patch Clamp). Individual cells may survive but this was not sufficient for our type of analysis.”

You wrote: Line 197, a typo "Activatiopn" needs to be corrected.

Response: Thanks for this comment. We have corrected the term.

Reviewer 3 Report

The manuscript entitle 'Induction of proteasome subunit LMP2 is required to induce active remodeling in cardiomyocytes' is well written and contains some general interests for the researcher who are interested in in vitro primary cardiac cell culture. Thus I believe the manuscript is worth to be published in 'medical sciences', if the authors could answer the questions below.

1st, The authors argued that p38 dependent LMP2 upregulation is prerequisite for ARVC in vitro adaptation. The LMP2 is a well known component of an immunoproteasome whose construction is stimulated by external stresses for instance, viral infection. Thus I wonder whether the expression level of the other components of the proteasome are also fluctuated during ARVC adaptation procedure. For example, LMP7, PA28 cap etc.

2nd, In Fig. 6C, the protein expression level of LMP2 doesn't seemd to be affected by the treatment of SB, considering the concomitant reduction of GAPDH. So the level of protein expression must be quantified in proportion to the control (GAPDH). 

Author Response

You wrote: The authors argued that p38 dependent LMP2 upregulation is prerequisite for ARVC in vitro adaptation. The LMP2 is a well known component of an immunoproteasome whose construction is stimulated by external stresses for instance, viral infection. Thus I wonder whether the expression level of the other components of the proteasome are also fluctuated during ARVC adaptation procedure. For example, LMP7, PA28 cap etc.

Response: Proteasome activity is regulated by inducible subunits that replace constitutive components of the proteasome complex, namely β isoforms. In cardiac tissue (and in isolated cardiomyocytes) protein expressions of the three inducible isoforms β1i (=Lmp-2), β2i, and β5i are low compared to immune cells. In principle they can be induced by cytokines. Whether cytokines or growth factors that are present in FCS are responsible for the observed effects is unclear. However, a change in proteasome activity requires the induction of these inducible isoforms, from which we identified Lmp-2 as relevant. Our siRNA experiments (Fig. 7) are in line with these assumptions that the relevant effect depends on Lmp-2. Therefore, no further isoforms were analyzed.

You wrote: In Fig. 6C, the protein expression level of LMP2 doesn't seemd to be affected by the treatment of SB, considering the concomitant reduction of GAPDH. So the level of protein expression must be quantified in proportion to the control (GAPDH).   

Response: We apologize for not having stated initially that all protein expression data are normalized to GAPDH; we now have added a sentence to the material and method section (2.4.). This was also the case in Fig. 6b. In the context of the study it is also important to note in Fig. 6B that no double band of Lmp-2 appears, indicating that the remaining Lmp-2 is not incorporated into proteasome complexes.